# Effect of the Metabotropic Glutamate Receptor Type 5 Negative Allosteric Modulator Dipraglurant on Motor and Non-Motor Symptoms of Parkinson’s Disease

**DOI:** 10.3390/cells12071004

**Published:** 2023-03-24

**Authors:** Mark P. Epping-Jordan, Françoise Girard, Anne-Sophie Bessis, Vincent Mutel, Christelle Boléa, Francis Derouet, Abdelhak Bessif, Brice Mingard, Stéphanie Barbier, Justine S. Paradis, Jean-Philippe Rocher, Robert Lütjens, Mikhail Kalinichev, Sonia Poli

**Affiliations:** Addex Therapeutics, S.A., Chemin des Mines 9, 1202 Geneva, Switzerland

**Keywords:** dipraglurant, mGlu5, depression, anxiety, obsessive-compulsive disorder, Parkinson’s disease

## Abstract

Parkinson’s disease (PD) patients suffer not only from the primary motor symptoms of the disease but also from a range of non-motor symptoms (NMS) that cause disability and low quality of life. Excessive glutamate activity in the basal ganglia resulting from degeneration of the nigrostriatal dopamine pathway has been implicated in the motor symptoms, NMS and dyskinesias in PD patients. In this study, we investigated the effects of a selective mGlu5 negative allosteric modulator (NAM), dipraglurant, in a rodent motor symptoms model of PD, but also in models of anxiety, depression and obsessive-compulsive disorder, all of which are among the most prevalent NMS symptoms. Dipraglurant is rapidly absorbed after oral administration, readily crosses the blood-brain barrier, and exhibits a high correlation between plasma concentration and efficacy in behavioral models. In vivo, dipraglurant dose-dependently reduced haloperidol-induced catalepsy, increased punished licks in the Vogel conflict-drinking model, decreased immobility time in the forced swim test, decreased the number of buried marbles in the marble-burying test, but had no effect on rotarod performance or locomotor activity. These findings suggest that dipraglurant may have benefits to address some of the highly problematic comorbid non-motor symptoms of PD, in addition to its antidyskinetic effect demonstrated in PD-LID patients.

## 1. Introduction

The diagnosis of Parkinson’s disease (PD) is based primarily on the classical motor symptoms of bradykinesia, rigidity, resting tremor and postural instability resulting from degeneration of the nigrostriatal dopaminergic pathway [1]. More recently, a wide range of non-motor symptoms (NMS) including psychiatric, cognitive, sleep, pain and autonomic disturbances has been recognized as an integral feature of PD. Most patients diagnosed with PD suffer from NMS, which can precede the onset of motor symptoms and are commonly present at the time of diagnosis [2,3]. These symptoms can worsen disability and greatly decrease the quality of life, sometimes to a greater extent than the motor complications of the disease [4]. Affective disorders are among the most common NMS in PD patients with an average reported prevalence for anxiety disorders of approximately 40% [5,6], while prevalence of depression ranges from 31 to 45%, depending on the diagnostic criteria [7,8], with a significant fraction exhibiting co-morbid anxiety and depression [9,10]. PD patients also experience a variety of compulsive disorders involving repetitive, stereotyped actions that are intended to decrease anxiety and avoid harm. Compulsive disorders in PD patients are associated with a higher levodopa (L-Dopa) dose or dopamine agonist treatment. They may be treated by adjusting the dopaminergic medication dose, although symptoms are often refractory requiring additional therapeutic intervention [4].

Loss of striatal dopamine input results in excessive glutamate activity in the basal ganglia contributing to the motor symptoms and complications of PD [11,12,13]. Glutamate acts through ionotropic and metabotropic receptors, mediating fast synaptic and slower modulatory signaling, respectively. Antagonists of ionotropic glutamate receptors have shown antiparkinsonian and antidyskinetic effects [14,15,16,17,18,19]; however, these drugs produce significant adverse effects, including dissociative states and cognitive impairment [17,20], that limit their clinical use.

Metabotropic glutamate receptors (mGlus) are a family of G protein-coupled receptors consisting of eight subtypes divided into three groups based on sequence homology, pharmacology and signal transduction mechanisms [21,22]. The metabotropic glutamate receptor subtype 5 (mGlu5) is highly expressed in medium spiny neurons and interneurons in the striatum and various output structures of the basal ganglia [23,24,25,26,27]. mGlu5 mediates excitatory postsynaptic activity and has a reciprocal, modulatory interaction with N-methyl-D-aspartate (NMDA) receptors [28,29]. Therefore, antagonism of mGlu5 may reduce excessive glutamate transmission throughout the basal ganglia structures implicated in the symptoms of PD, while avoiding adverse effects of ionotropic glutamate receptor antagonists. 

A large body of evidence supports a role for inhibition of mGlu5 in the treatment of motor symptoms and psychiatric NMS of PD. Negative allosteric modulators (NAMs) of mGlu5 exhibit antiparkinsonian effects [30,31,32,33] and reverse levodopa-induced dyskinesia (LID) in rodent [34,35,36] and primate models [37,38,39], as well as in recent human clinical trials [40,41]. In addition, mGlu5 NAMs exhibit anxiolytic-, antidepressant- and anticompulsive-like effects in preclinical models [33,42,43,44,45]. In addition, mGlu5 NAMs reduce pain in a range of rodent models [46]. These results suggest that mGlu5 NAMs may be useful in the treatment of the motor and non-motor symptoms, as well as the dyskinesia resulting from the loss of dopaminergic neurons and long-term L-Dopa treatment. 

Dipraglurant (ADX48621) is a potent and highly selective mGlu5 NAM [47] originating from a high throughput screening campaign of Addex’s small molecule library and optimized through intensive medicinal chemistry efforts. Dipraglurant has successfully completed a Phase IIa clinical trial in levodopa-induced dyskinesia [41] and further clinical development is planned.

The present study examined the effects of dipraglurant in rodent models of PD, anxiety, depression and obsessive-compulsive disorder (OCD). 

## 2. Materials and Methods

### 2.1. Subjects

Subjects were adult male C57Bl/6J mice (Charles River, Saint-Germain-Nuelles, France), Sprague-Dawley rats (280–430 g; Charles River, Saint-Germain-Nuelles, France) and Wistar (Han) rats (180–250 g; Janvier, Le Genest-Saint-Isle, France). We used C57Bl/6J mice together with Sprague-Dawley and Wistar rats based on a large body of internal data in corresponding models where the activity of positive and negative controls are robust and well-reproducible. Upon arrival to the animal facility, mice were group-housed (5 per cage) in Makrolon type II cages (16 × 22 × 14 cm), whereas rats were group-housed (2 per cage) in Makrolon type III cages (22 × 37 × 18 cm). Animals were maintained on a 12 h light/dark cycle (lights on from 7:00am to 7:00pm) under constant temperature (22 ± 2 °C) and humidity (>45%) conditions. Food (SDS-Dietex, Saint-Gratien, France) and water were available ad libitum, unless temporary water-deprivation was required by the experimental procedure (see below). Animals were acclimated for at least 10 days before experimentation. All experimental procedures were approved by the Ethical Committee of Addex Therapeutics and performed in full compliance with the European Communities Council Directive of 24 November 1986 (86/609/EEC). 

### 2.2. Drugs

Dipraglurant (ADX48621, 6-fluoro-2-(4-pyridin-2-ylbut-3-ynyl)imidazo [1,2-*a*]pyridine) monophosphate salt and MTEP (3-((2-Methyl-4-thiazolyl)ethynyl)pyridine) were synthetized at Addex Therapeutics. Chlordiazepoxide hydrochloride, desipramine hydrochloride and imipramine hydrochloride were purchased from Sigma-Aldrich (Buchs, Switzerland). Diazepam was obtained from Coopération Pharmaceutique Française (Paris, France). Haloperidol was purchased from Janssen-Cilag (Issy-Les-Moulineaux, France).

Dipraglurant, chlordiazepoxide and imipramine were dissolved in distilled water. Haloperidol and desipramine were dissolved in saline. MTEP was dissolved in 20% Tween-80 in 0.1N hydrochloric acid (HCl). The MTEP solution was subsequently adjusted to pH 6–7 with 1M sodium bicarbonate (NaHCO_3_) solution. Diazepam was suspended in 0.2% hydroxypropyl methylcellulose (CMC) in distilled water. Solutions or suspensions were administered at 10 mL/kg volume in mice, and in 3 or 5 mL/kg volume in rats. All solutions and suspensions were prepared fresh daily. All doses are expressed as the free base. 

### 2.3. Plasma and Cerebrospinal Fluid Sampling

Blood sampling after pharmacodynamic assessment is described in the relevant sections. Blood sampling for pharmacokinetic assessment was performed as follows: rats were surgically implanted under isoflurane–oxygen anesthesia (1–2%) with jugular catheters which ran subcutaneously from the jugular vein and exited in the midscapular region. Animals were allowed to recover for 2–6 days following surgery before administration of dipraglurant. CSF was sampled as previously described [48]. Rats received oral gavage of 10 mg/kg dipraglurant. Serial blood and CSF samples were collected from the catheter at 0.25, 0.5, 1, 2, 4, 6, 8 and 24 h following administration. Blood samples were collected in 1.5 mL polyethylene Eppendorf tubes containing 4 µL of 15% EDTA solution and immediately placed on ice. Samples were centrifuged at 4 °C for 12 min at 5900× *g*. Plasma was transferred to 1.5 mL Eppendorf tubes and stored at −20 °C until analysis. CSF was collected in 20 µL heparin-coated capillary tubes. For sample preparation, CSF was transferred into Eppendorf tubes containing 20 µL of control rat plasma. Samples were checked carefully for blood contamination. Subsequently, samples were placed on ice and, if necessary, kept frozen at −20 °C until analysis. CSF sample extraction was identical to the procedure described for plasma samples (see below).

### 2.4. Bioanalytical Method

To precipitate proteins, 150 µL of acetonitrile was added to 50 µL of plasma spiked with 10 µL DMSO for unknown samples or 10 µL of dipraglurant for calibration and Quality Control (QC) samples. After vortexing and centrifugation (15 min, 4 °C, 13,200 rpm), a portion (100 µL) was transferred into a 384-well analytical plate. A total of 5 µL of the supernatant was injected into a UPLC system (Waters, Milford, USA) coupled with a mass spectrometer API 3200 (Applied Biosystems, Les Ulis, France). Samples were injected onto an Acquity BEH C_18_ reverse phase column. Elution was performed with a high-pressure linear gradient from 25 to 100% acetonitrile in 10 mM ammonium formate at pH 3.5. Run time was 1.5 min (retention time = 0.43 min). The electrospray positive ionization was used in MRM mode (transition 273.04/132.2). Two calibration curves and three QC levels in duplicate were used to quantify and validate the run (Quadratic 1/x).

### 2.5. Plasma Protein Binding 

Plasma protein binding was measured by equilibrium dialysis using 96-well plates specifically designed for this purpose (HT Dialysis, Gales Ferry, CT, USA). This reusable 96-well plate is assembled so that each well is divided vertically in two parts by a dialysis membrane. MWCO regenerated dialysis cellulose membranes (Dialysis Membrane Strips, HT Dialysis, Gales Ferry, CT, USA) with a molecular weight cut-off of 12–14,000 Da were conditioned according to the manufacturer’s instructions and used for all experiments. A total of 1 µL of a 1 mg/mL DMSO solution of dipraglurant was added to rat, mouse or human plasma to reach the final concentration of 1000 ng/mL. An amount of 150 µL of the resulting plasma solution was added on one side of the well and, simultaneously, the pH 7.4 phosphate buffer solution was added to the other side of the well. Experiments were carried out in duplicate for each species and each time point. Individual wells were used for each time point. The plates were sealed and set in an incubator at 37 °C under gentle shaking. Samples were taken from the plasma and buffer compartments at the start of the experiments and after 6 and 7 h and were immediately stored at 4 °C. 

The plasma and buffer samples were subsequently analyzed by a specific LC-MS method to determine dipraglurant concentrations. The portion of bound dipraglurant in plasma was calculated according to the following equation:% NCE bound=(Cplasma − Cbuffer)×100/Cplasma
where *C_plasma_* is the total (bound + free) concentration of dipraglurant in plasma after equilibrium is reached, and *C_buffer_* is the concentration of dipraglurant measured in the buffer solution at the same time. This equation is valid only when equilibration between the plasma and buffer solutions is completed. Previous experiments demonstrated that, for the reference substances, equilibrium between the plasma and buffer solution is reached after 5–6 h.

### 2.6. Haloperidol-Induced Catalepsy

Sprague-Dawley rats (*n* = 10/group) were treated interperitoneally (i.p.) with haloperidol (1 mg/kg). Thirty minutes later, animals were treated with either dipraglurant (3, 10 and 30 mg/kg, orally p.o.), vehicle (distilled water, p.o.) or MTEP (30 mg/kg, i.p.). Sixty minutes later (i.e., ninety minutes following haloperidol administration), animals were placed, with all four paws, on a vertical wire grid, with the head pointing toward the ceiling. Moving both forepaws to relocate the body determined the duration of catalepsy (s) with a “cut-off” time of 120 s. Blood samples were taken from all dipraglurant-treated animals at the end of the experiment (i.e., 65–70 min post-dosing) and dipraglurant plasma concentrations were determined. 

### 2.7. Vogel Conflict-Drinking Test in Rats

Sprague-Dawley rats, water-deprived for 48 h, were placed in a Vogel chamber in which a mild shock was delivered via the spout of the drinking bottle each time animals performed 20 licks. The test session was preceded by 2 habituation sessions, performed 24 h apart. In the first session, 48 h before the test, animals were placed in Vogel chambers in pairs, for 10 min. At the end of the habituation session animals were returned to their home-cages and water bottles were removed for the first 24 h water deprivation period. In the second session, 24 h before the test, animals were placed individually in Vogel chambers and allowed to drink freely for 4 min after the first lick. The number of licks was monitored and animals that performed fewer than 50 licks were excluded from this study. Animals were then returned to their home cages and were allowed to drink freely for 1 h. At the end of this period water bottles were removed from the cages for the second 24 h water deprivation period.

On the testing day, rats (*n* = 8–10/group) were treated with either dipraglurant (3, 10, 30 mg/kg, p.o.), vehicle (distilled water, p.o.) or chlordiazepoxide (30 mg/kg, i.p.). Sixty minutes later, the animals were individually placed in Vogel chambers for five minutes. During this period an electric shock (0.6 mA, 0.5 s) was delivered for every 20 licks performed. The total number of licks and the latency (s) to perform the first lick were recorded. Blood samples were taken from all dipraglurant-treated animals at the end of the experiment (i.e., 65–70 min post-dosing) and dipraglurant plasma concentration was determined. 

### 2.8. Marble-Burying Test in Mice

C57Bl6/J mice (*n* = 10/group) were treated with either dipraglurant (10, 30, 50 mg/kg, p.o.), vehicle (distilled water, p.o.) or chlordiazepoxide (30 mg/kg, i.p.). Thirty minutes later, animals were placed individually into clear Makrolon type II cages, with clear Plexiglas covers, containing 5 cm of sawdust bedding (Lignocel, Harlan Laboratory, Loughborough, UK) and ten marbles evenly spaced against the walls. The mice were left undisturbed in the cages for 30 min. At the end of the experiment animals were removed from the test cages and the number of buried marbles was counted.

Blood samples were taken from all dipraglurant-treated animals at the end of the experiment (i.e., 65–70 min post-dosing) and dipraglurant plasma concentrations were determined. 

### 2.9. Forced Swim Test 

#### 2.9.1. Mice

C57Bl6/J mice were pre-exposed to water for 15 min in a training session one day before the experiment. They were placed individually into clear glass cylinders (height: 25 cm, diameter: 10 cm) containing water 10 cm deep at 24 ± 1 °C. The water was changed between subjects. At the end of each swimming session, mice were dried with a paper-towel and kept under a heat lamp for approximately 10 min before being returned to their home cages. On the test day, 24 h later, animals (*n* = 10/group) were treated with either dipraglurant (30, 50 mg/kg, p.o.), vehicle (distilled water, p.o.), or desipramine (10 mg/kg, i.p.). Sixty minutes after drug treatment, animals were exposed to a test swim session for six minutes under conditions identical to those used during training. Blood samples were taken from all dipraglurant-treated animals at the end of the experiment (i.e., 65–70 min post-dosing) and dipraglurant plasma concentration was determined. 

#### 2.9.2. Rats

Wistar rats were pre-exposed to water in a training session one day before the experiment. They were placed individually into clear glass cylinders (height: 46 cm, diameter: 20 cm) containing water 30 cm deep at 25 ± 1 °C for 15 min. The water was changed between subjects. At the end of each swimming session rats were dried with a paper-towel and kept under a heat lamp for approximately 15 min before being returned to their home cages. On the test day, 24 h later, animals (*n* = 10/group) were treated with either dipraglurant (3, 10, 30 mg/kg, p.o.), vehicle (distilled water, p.o.) or imipramine (64 mg/kg, i.p.) three times, 24 h, 4 h and 1 h before the test. Animals were tested in the swim session for 5 min under conditions identical to those used during training. 

All forced swim test sessions were recorded by a video camera positioned on the side of the cylinder. A trained observer blind to the treatment scored the tapes. The behavioral measures assessed included the duration of immobility (s) exhibited during the test. An animal was considered immobile if it remained floating motionless in the water making only the movements necessary to keep its head above the water. 

### 2.10. Rotarod Test 

#### 2.10.1. Mice

A mouse rotarod apparatus (MED Associates; St. Albans, VT, USA) with a constant speed (16 rotations per min) was used in this experiment. The rotarod was turned on at least 30 min before starting the experiment in order to habituate the animals to the noise generated by the apparatus. The day before the experiment C57Bl6/J mice received 3 training sessions (3 min per session) performed 60 min apart to familiarize the animals with the apparatus. Animals that fell off the rotarod in 2 out of 3 training sessions were excluded from the experiment. On the day of the experiment animals were treated p.o. with either dipraglurant (10, 30, 50 mg/kg), vehicle (distilled water) or chlordiazepoxide (30 mg/kg) and 60 min later were tested on the rotarod (cut-off time of 180 s). 

#### 2.10.2. Rats

A rat rotarod apparatus (Ugo Basile 7700, Comerio, Italy) with a constant speed (15 rotations per minute) was used in this experiment. The rotarod was turned on at least 30 min before the start of the experiment to habituate the animals to the noise generated by the apparatus. Sprague-Dawley rats were habituated to the rotarod 24 h before test. One hour before the test, animals received 6 training sessions performed 4–10 min apart. In each training session the time that animals remained on the rod was recorded (with a cut-off time of 180 s). Between each trial, animals were kept in their home cages. Animals that were able to stay on the rod without falling for the entire 180 s session for 3 out of 6 trials were selected for this study. For the test, animals were treated with either dipraglurant (3, 10, 30 mg/kg, p.o.), vehicle (water, p.o.) or diazepam (8 mg/kg, i.p.). Sixty minutes later, animals were placed on the rotating rod under conditions identical to the training session and the time (s) each animal remained on the rod was recorded. 

### 2.11. Locomotor Activity 

#### 2.11.1. Mice

In C57Bl6/J mice spontaneous locomotor activity, assessed as horizontal distance travelled (cm), was monitored using eight opaque Plexiglas arenas (35 × 35 × 50 cm) with a computerized video tracking analysis system (Viewpoint; Lyon, France). Animals (*n* = 8/group) were treated with either dipraglurant (10, 30, 50 mg/kg, p.o.), vehicle (distilled water, p.o.) or chlordiazepoxide (10 mg/kg, i.p.). Sixty minutes later, animals were placed individually into arenas and their locomotor activity was monitored for sixty minutes. 

#### 2.11.2. Rats

In Sprague-Dawley rats spontaneous locomotor activity, assessed as horizontal distance travelled (cm), was monitored using eight opaque Plexiglas arenas (50 × 50 ×50 cm) with a computerized video tracking analysis system (Viewpoint; Lyon, France). Animals (*n* = 8/group) were treated with either dipraglurant (3, 10, 30 mg/kg, p.o.), vehicle (distilled water p.o.) or chlordiazepoxide (50 mg/kg, i.p.). Sixty minutes later, animals were placed individually into arenas and their locomotor activity was monitored for sixty minutes.

### 2.12. Data Analysis

Time spent on the vertical grid (s; haloperidol-induced catalepsy test in rats), number of punished licks, latency (s) to commence licking (Vogel Test in rats), immobility time (Forced Swim Test in mice and rats), number of buried marbles (Marble Burying Test in mice), total distance traveled (Locomotor Activity Test in mice and rats) and time spent on the rotarod (Rotarod Test in mice and rats) were analyzed by one-way ANOVA, followed by planned comparisons. The alpha level chosen was *p* < 0.05. Data from the behavioral experiments were analyzed using GraphPad Prism 9 (GraphPad Software Inc., San Diego, CA, USA). 

## 3. Results

### 3.1. Pharmacokinetic Profile in Rats

Dipraglurant was rapidly absorbed following p.o. administration in rats reaching maximum concentration (C_max_) in both plasma and CSF 0.5 h after dosing (Figure 1). The CSF concentration time course closely followed the plasma concentration (Figure 1). The CSF/plasma ratio was 0.015. The unbound fraction was determined to be 1.9% in rat and 9.1% in mouse plasma. 

### 3.2. Dipraglurant Shows Antiparkinsonian Effect in Haloperidol-Induced Catalepsy Test

Dipraglurant (1, 3, 10 and 30 mg/kg p.o.) dose-dependently reduced the time animals stayed on the vertical grid in the rat catalepsy test [F(4,55) = 10.21, *p* < 0.001] (Figure 2A). Specifically, a trend of activity at 1 mg/kg was followed by 40% (*p* < 0.01), 53% and 65% (both *p* < 0.001) reductions in time spent on the grid at 3, 10 and 30 mg/kg of dipraglurant, respectively (Figure 2A). The magnitude of the effect seen at 30 mg/kg dipraglurant was similar to the one measured in MTEP-treated animals (64%, *p* < 0.001; Figure 2A). The corresponding mean plasma concentration of dipraglurant in animals treated at 1, 3, 10 and 30 mg/kg were 54, 328, 1727 and 3617 ng/mL (Table 1). There was a high correlation between plasma concentration and efficacy (Figure 2B). The maximum activity of dipraglurant in the catalepsy test was reached at plasma concentrations of ≥1000 ng/mL (Figure 2B).

### 3.3. Anxiolytic Effect of Dipraglurant in the Vogel Test

Dipraglurant (3, 10 and 30 mg/kg p.o.) dose-dependently increased the number of punished licks in the rat Vogel test [F(4, 42) = 3.48; *p* < 0.05] (Figure 3). Specifically, a mild trend of activity at 3 mg/kg, was followed by 2.2- and 2.5-fold increases (both *p* < 0.05) in punished licks at 10 and 30 mg/kg, respectively (Figure 3). The latency to commence licking (sec) was not affected by either dipraglurant or chlordiazepoxide (data not shown). The corresponding mean plasma concentration of dipraglurant in animals treated at 3, 10 and 30 mg/kg were 501, 1120 and 2043 ng/mL (Table 1).

### 3.4. Anticompulsive Effect of Dipraglurant in the Marble Burying Test

Dipraglurant (10, 30 and 50 mg/kg, p.o.) dose-dependently reduced the number of buried marbles in the mouse marble burying test [F(4, 45) = 24.2; *p* < 0.001] (Figure 4). Specifically, a lack of activity at 10 mg/kg, was followed by 32% (*p* < 0.01) and 78% (*p* < 0.001) reductions in burying marbles at 30 and 50 mg/kg, respectively (Figure 4). The magnitude of the effect seen at 50 mg/kg dipraglurant was similar to that seen in chlordiazepoxide-treated animals (72%, *p* < 0.001; Figure 4). The corresponding mean plasma concentration of dipraglurant in animals treated at 10, 30 and 50 mg/kg were 26, 212 and 865 ng/mL (Table 1). 

### 3.5. Antidepressant Effect of Dipraglurant in the Forced Swim Test

Dipraglurant (30 and 50 mg/kg p.o.) dose-dependently reduced the duration of immobility in the mouse FST [F(3, 36) = 11.57; *p* < 0.001], by 30% (*p* < 0.01) and 40% (*p* < 0.001) at 30 and 50 mg/kg, respectively (Table 2). Dipraglurant (3, 10 and 30 mg/kg, p.o.) also dose-dependently reduced immobility time in the rat FST (Table 2). Oral administration of 3 and 10 mg/kg resulted in ~15% reduction in immobility time (*p* < 0.05), and ~20% reduction (*p* < 0.01) at 30 mg/kg (Table 2). Imipramine-treated animals exhibited ~50% reduction (*p* < 0.001) in immobility (Table 2). 

### 3.6. Dipraglurant Shows No Motor Impairment, as Measured in the Rotarod Test 

Mice treated with dipraglurant (10, 30 and 50 mg/kg p.o.) did not differ from vehicle-treated animals in the time spent on the rotarod. In contrast, chlordiazepoxide-treated mice spent markedly less time (*p* < 0.001) on the rotarod (Table 3).

Rats treated with dipraglurant (3, 10 and 30 mg/kg p.o.) also did not differ from vehicle-treated controls in the time spent on the rotarod. In contrast, diazepam-treated rats spent markedly less time (*p* < 0.001) on the rotarod (Table 3).

### 3.7. No Effect of Dipraglurant on Spontaneous Locomotor Activity 

Mice treated with dipraglurant (10, 30 and 50 mg/kg p.o.) did not differ from vehicle- or chlordiazepoxide-treated animals in their locomotor activity profiles (Figure 5A) and in the total distance traveled (Figure 5B). Rats treated with dipraglurant (3, 10 and 30 mg/kg p.o.) also did not differ from vehicle-treated controls in their locomotor activity profiles (Figure 5C) and in the total distance traveled (Figure 5D). In contrast, chlordiazepoxide significantly (*p* < 0.001) suppressed locomotor activity in rats (Figure 5D).

## 4. Discussion

There is a significant unmet medical need for medications to treat the full range of symptoms and motor complications of PD. The initial motor symptoms are treated effectively with dopamine replacement therapies, but as the disease progresses, dyskinesias inevitably appear in the vast majority of patients [49,50]. In addition, PD patients experience a wide range of NMS at rates higher than those seen in the general population [2,4,9]. Excessive glutamate activity has been implicated in the motor symptoms and dyskinesias as well as in the affective and compulsive disorders frequently observed in PD patients [49,51]. Antagonists of ionotropic glutamate receptors improve PD symptoms and dyskinesias confirming the therapeutic utility of reducing glutamate transmission; however, unacceptable side effects of these agents limit their clinical usefulness [17,20,52,53]. Compounds acting at mGlu receptors have received increasing attention as potential treatments for PD due to their effectiveness in animal models and superior side effect profile. Here, we report that the mGlu5 NAM dipraglurant shows antiparkinsonian, anxiolytic-, antidepressant- and anticompulsive-like effects in preclinical models. 

Dipraglurant is a potent and selective mGlu5 NAM. In vitro, its concentrations producing half-maximal inhibition of the glutamate response (IC_50_ values) on recombinant mGlu5 receptors were 21 ± 1 nM and 45 ± 2 nM on human mGlu5 [47]). Dipraglurant has a good oral bioavailability and excellent brain-penetration-achieving CSF concentrations comparable to the plasma-free fraction. The compound is absorbed rapidly following oral administration reaching peak plasma concentrations in approximately 30 min. 

In the present experiment, dipraglurant (10 and 30 mg/kg p.o.) reversed catalepsy induced by haloperidol with a maximum effect equivalent to the mGlu5 NAM tool compound MTEP. These results are consistent with previous studies of the mGlu5 antagonists MPEP and MTEP [31,34], although one study [54] reported no effect of acute or chronic MPEP. Antiparkinsonian effects of glutamate antagonists have been observed consistently in models employing acute haloperidol treatment [31,34,55,56,57]. However, antiparkinsonian effects of mGlu5 antagonists administered in the absence of L-Dopa have not been observed in rat 6-OHDA-lesioned or MPTP-treated monkeys rendered dyskinetic by chronic L-Dopa treatment [37,38,39,58]. These manipulations typically produce extensive lesions associated with more advanced stages of PD in humans [58,59]. Nevertheless, in a recent clinical trial of levodopa-induced dyskinesias, the mGlu5 inhibitor AFQ056 (mavoglurant) exhibited antiparkinsonian effects greater than L-Dopa alone on two individual study days [40]. Patients in this study had been on L-Dopa for at least 3 years and already had developed significant dyskinesias, suggesting extensive degeneration of the nigrostriatal pathway. Under these circumstances, the antiparkinsonian potential of mGlu5 receptor blockade may be obscured by the severity of motor symptoms and the presence of profound dyskinesia. Moreover, antiparkinsonian effects may be masked in human studies in which subjects are treated with an optimal dose of L-Dopa [14]. Thus, it is possible that mGlu5 NAMs may exert antiparkinsonian effects as a monotherapy especially in early PD prior to extensive nigrostriatal degeneration, or they may prove useful as an adjunct to dopamine replacement therapies, allowing for lower doses of dopaminergic drugs, which may delay or prevent the development of dyskinesias known to be related to dose and duration of treatment [49]. In addition, several studies have revealed neuroprotective effects of genetic deletion of mGlu5 or mGlu5 antagonists in 6-OHDA- or MPTP-treated animals [60,61,62,63,64,65,66,67] suggesting that antagonism of mGlu5 may retard further neurodegeneration and improve motor symptoms in early PD. Long-term mGlu5 inhibition in patients appears safe and well tolerated [41,68,69], warranting further clinical exploration to demonstrate such effects in PD patients.

Psychiatric disorders are the most frequently observed NMS in PD patients, may persist during both medication “on” and “off” states and often contribute to greater disability and poorer quality of life than motor symptoms [2,4,9]. Non-motor symptoms may be related to the pathophysiology of PD, or they may be a consequence of medications used to treat motor symptoms. Recognition and adequate treatment of NMS is key to successful treatment of PD patients [4]. In the present study, dipraglurant (10 and 30 mg/kg p.o.) had a significant anxiolytic-like effect in the Vogel conflict-drinking model comparable in magnitude to chlordiazepoxide but without a corresponding effect on spontaneous locomotor activity. Our results are consistent with previous studies using mGlu5 NAM tool compounds [42,70,71,72,73]. Anxiety disorders are observed in approximately 40% of PD patients, often precede the appearance of motor symptoms [2,3] and are strongly associated with a negative impact on quality of life [3]. The etiology of anxiety disorders in PD patients is unknown and likely multifactorial; however, a recent study demonstrated that partial 6-OHDA lesions of the dorsal striatum induced anxiety-like behaviors in rats which were associated with decreased firing in putative glutamatergic afferent projection neurons in the basolateral amygdala (BLA) [3]. Chronic MPEP treatment reversed the anxiogenic-like effect and restored normal BLA firing rates suggesting that partial dopamine denervation of the striatum, intended to mimic the prodromal stage of PD, may result in anxiety which is reversible with antagonism of mGlu5. Taken together with our data, these results suggest that mGlu5 NAMs may be particularly effective in treating anxiety in PD patients.

Antidepressant-like activity in the forced swim test was observed at all dipraglurant doses (3–30 mg/kg po) in the absence of effects on rotarod performance or locomotor activity. Similar results have been reported for the tool compounds MPEP and MTEP in the forced swim test in rats [44] and mice [74,75] and are confirmed to be due to activity at mGlu5 in knockout mice [74]. MPEP also has been reported to exert antidepressant-like effects in the olfactory bulbectomy model of depression in rats [76]. The prevalence of depression in PD patients typically ranges from 31 to45% but has been reported in up to 72% of PD patients [4,7,8]. Patients with depression and PD exhibit less sorrow and self-blame but more anxiety, irritability, cognitive impairment and suicidal ideation than depressed non-PD patients [4]. This evidence suggests that the pathophysiology of depression in PD patients, which may differ from non-PD patients, may be linked to excessive glutamate transmission, and thus these patients may benefit from a treatment with drugs that reduce glutamate signaling.

Anticompulsive-like effects of dipraglurant were observed in the marble-burying task in mice at doses (30 and 50 mg/kg p.o.) which had no effect on spontaneous locomotor activity or rotarod performance. Previous studies have reported that MPEP decreased marble burying in normal mice [33,77], mice treated prenatally with valproic acid as a model for autism [78] and FmR1 knockout mice generated as a model for fragile X syndrome [45]. Accumulating evidence indicates that excessive glutamate activity is involved in OCD including elevated CSF glutamate levels in drug-naïve patients [79], improvement of symptoms following treatment with glutamate antagonists [51,80,81] and increased glutamatergic activity in the cortico-striato-thalamo-cortical (CSTC) circuit [82]. This last observation is particularly interesting because it suggests that OCD and compulsive disorders in PD may share a common pathophysiological substrate of excessive glutamate activity in the CSTC circuit. Along with our results, these data support the potential for mGlu5 NAMs in the treatment of compulsive disorders in PD.

The ED_50_ values of dipraglurant obtained in the current series of experiments differed across models and species. For example, the ED_50_ of dipraglurant in the rat model of catalepsy (2.83 mg/kg) was markedly lower than the value obtained in the mouse marble burying test (43 mg/kg). Most likely, these are linked to the differences in mGlu5 receptor distribution in neural circuits mediating a particular behavioral response, as well as species-specific differences in mGlu5 receptors. 

In summary, the mGlu5 NAM dipraglurant is effective in rodent models of parkinsonism, anxiety, depression and compulsive disorders. The present results support the hypothesis that excessive glutamate activity contributes to various symptoms of PD. An initial clinical trial with mGlu5 NAM mavoglurant showed that inhibition of mGlu5 is effective in reducing LID in PD patients [40] and dipraglurant has successfully completed a Phase IIa trial in PD-LID [41]. Additional studies with mavoglurant in PD-LID patients showed no benefit of the treatment [83], however this may be due to insufficient exposure, and therefore mGlu5 inhibition, at the time patients were experiencing peak dyskinesia. Thus, blocking mGlu5 just before or while L-dopa levels peak in the plasma and the brain may provide the most significant anti-dyskinetic effect in PD-LID patients. 

Negative allosteric modulators of mGlu5, and dipraglurant in particular, represent an attractive approach to treat the primary motor symptoms of Parkinson’s disease, concomitant psychiatric non-motor symptoms and dyskinesias induced by long-term L-Dopa treatment. 

## Figures and Tables

**Figure 1 cells-12-01004-f001:**
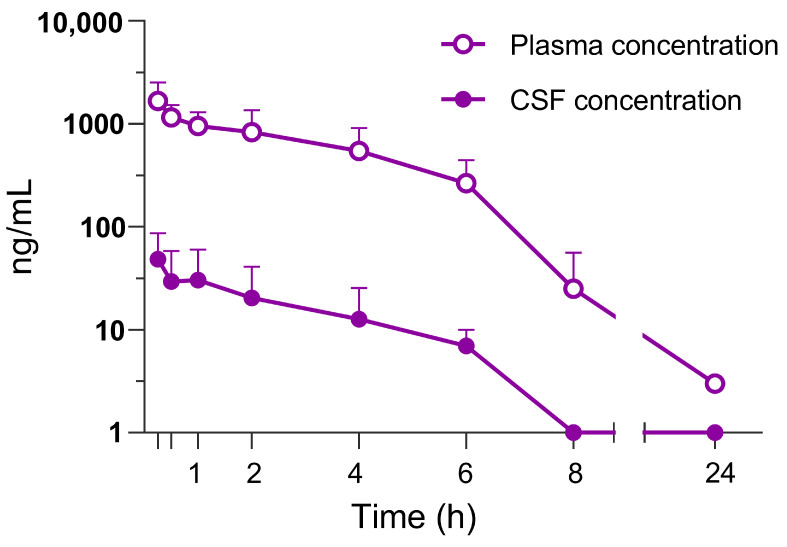
Plasma and CSF concentrations of dipraglurant after oral administration of 10 mg/kg of the compound administered at 3 mL/kg in water. Sprague-Dawley rats (*n* = 5) were used to collect blood via jugular vein catheter for standard plasma preparation at 0.25, 0.5, 1, 2, 4, 6, 8 and 24 h. Simultaneously, 8 CSF samples were collected at the same time points. Each point represents the measured mean ± SD.

**Figure 2 cells-12-01004-f002:**
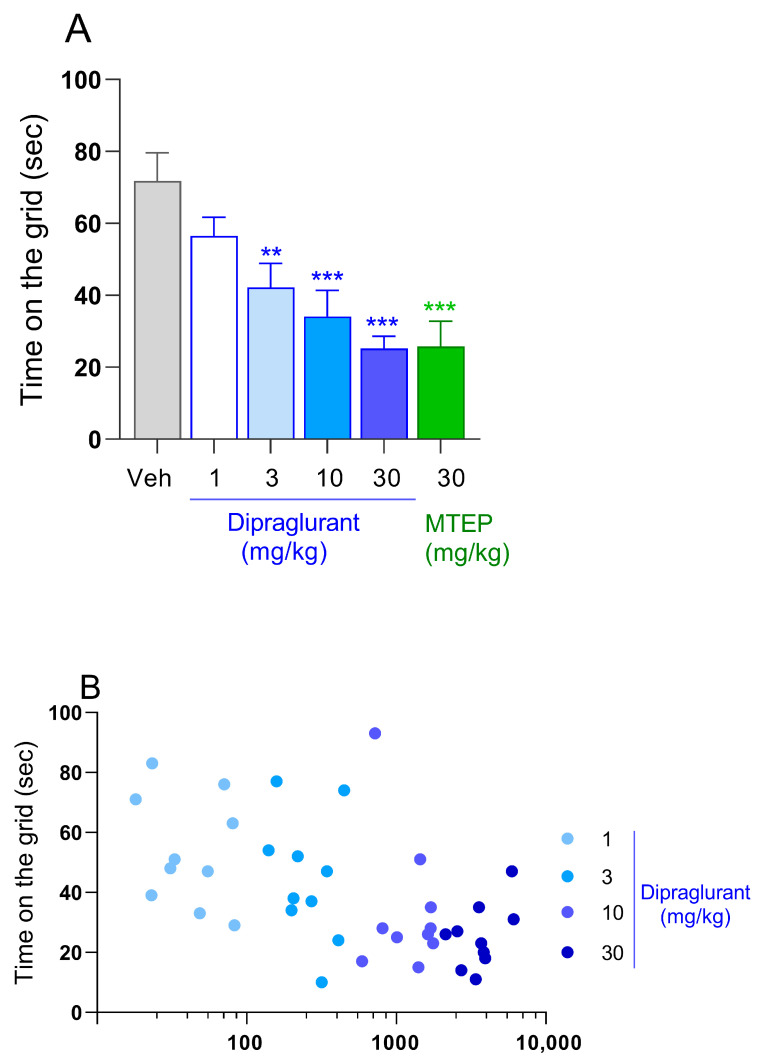
(**A**) Time spent by Sprague-Dawley rats (*n* = 10/group) on the vertical grid during the haloperidol-induced catalepsy test. Animals were pretreated with haloperidol (1 mg/kg, i.p.) before receiving either dipraglurant (1, 3, 10, 30 mg/kg p.o.), vehicle (distilled water, p.o.) or MTEP (30 mg/kg, i.p.) and then placed on the grid. Each point represents the observed mean ± SEM, ** *p* < 0.01, *** *p* < 0.001 compared to Vehicle (Veh). (**B**) Pharmacokinetic-pharmacodynamic relation between plasma concentration of dipraglurant and time on the grid. Each point represents individual animals.

**Figure 3 cells-12-01004-f003:**
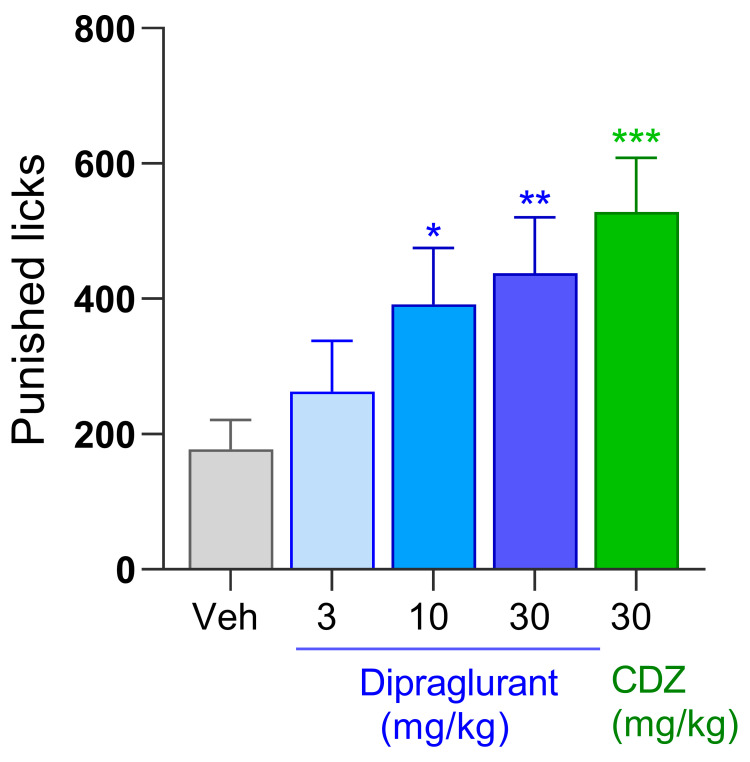
Number of punished licks made by Sprague-Dawley rats (*n* = 8–10/group) during the Vogel test. Animals were treated with either dipraglurant (3, 10, 30 mg/kg p.o.), vehicle (distilled water, p.o.) or chlordiazepoxide (CDZ; 30 mg/kg, i.p.) Each point represents the observed mean ± SEM, * *p* < 0.05, ** *p* < 0.01, *** *p* < 0.001 compared to vehicle (Veh).

**Figure 4 cells-12-01004-f004:**
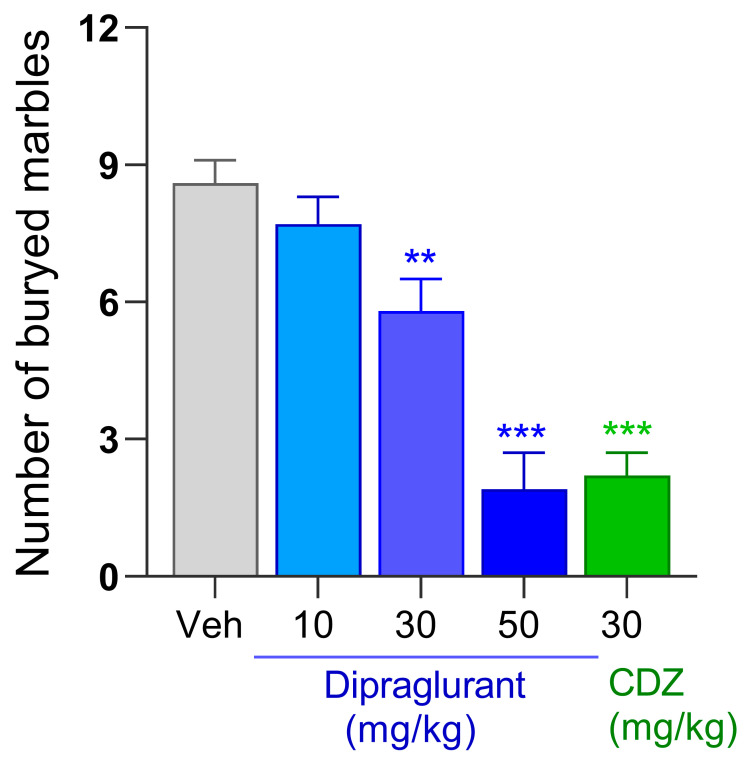
Number of buried marbles by C57Bl6/J mice (*n* = 10/group) during the marble burying test. Animals were treated with either dipraglurant (10, 30, 50 mg/kg p.o.), vehicle (distilled water, p.o.) or chlordiazepoxide (CDZ; 30 mg/kg, i.p.). Each point represents the observed mean ± SEM, ** *p* < 0.01, *** *p* < 0.001 compared to vehicle (Veh).

**Figure 5 cells-12-01004-f005:**
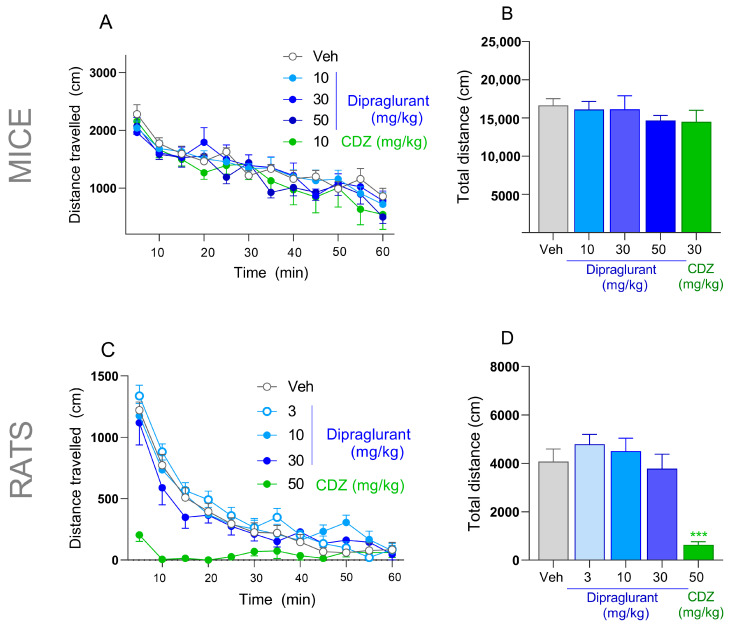
(**A**,**B**) Locomotor activity of male C57Bl6/J mice (*n* = 8/group) and (**C**,**D**) Sprague-Dawley rats (*n* = 8/group) during 60 min, expressed as time-course profiles (**A**,**C**) or total traveled distances (**B**,**D**). Mice were treated with either dipraglurant (10, 30, 50 mg/kg p.o.), vehicle (distilled water, p.o.) or chlordiazepoxide (CDZ; 10 mg/kg, i.p.). Rats were treated with either dipraglurant (3, 10, 30 mg/kg p.o.), vehicle (distilled water, p.o.), or CDZ (50 mg/kg, i.p.) Each point represents the observed mean ± SEM, *** *p* < 0.001 compared to vehicle (Veh).

**Table 1 cells-12-01004-t001:** Pharmacokinetic-pharmacodynamic relationship between plasma and CSF concentrations of dipraglurant and its efficacy in in vivo models. Pharmacological effect per dose indicated as * *p* < 0.05, ** *p* < 0.01, *** *p* < 0.001.

Domain	Model	Doses (mg/kg)	Plasma Conc. (ng/mL)	Calculated CSF Conc. (ng/mL)	Calculated CSF Conc. (nM)	CSF/IC_50_	ED_50_ (mg/kg)
Learned anxiety	Vogel conflict drinking test (rat)	3	501 ± 97	9.5	25.6	0.54	7.4 ± 0.9
		10 *	1120 ± 148	21.3	57.5	1.22	
		30 **	2043 ± 420	38.8	104.8	2.23	
Innate anxiety (OCD–like)	Marble burying (mouse)	10	26 ± 5	2.4	6.5	0.14	43 ± 6
		30 *	212 ± 28	19.3	52.1	1.11	
		50 ***	865 ± 156	78.7	212.5	4.52	
Depression	Forced swim test (mouse)	30 **	946 ± 264	86.1	232.5	4.95	~30
		50 ***	2266 ± 627	206.2	556.7	11.84	
	Forced swim test (rat)	3 **, 10 *, 30 **	N.D.				N.D.
Parkinson’s disease	Haloperidol-induced catalepsy (rat)	1	54 ± 10	1.0	2.7	0.06	2.83 ± 0.3
		3	328 ± 25	6.2	16.7	0.35	
		10 **	1723 ± 218	32.7	88.3	1.88	
		30 ***	3617 ± 383	68.7	185.5	3.95	

**Table 2 cells-12-01004-t002:** Effect of dipraglurant in the forced swim test (FST) in C57Bl6/J mice (administered at 30, 50 mg/kg) and Wistar rats (administered at 3, 10, 30 mg/kg), in comparison with vehicle, desipramine (in mice) or imipramine (in rats). Results are expressed as mean ± SEM, * *p* < 0.05, ** *p* < 0.01, *** *p* < 0.001 compared to vehicle (Veh).

Species	Compound	Administration Route	Dose (mg/kg)	Immobility Duration (s) Mean ± S.E.M.
Mouse	Dipraglurant	p.o.	0	188.2 ± 9.5
			30	134.4 ± 16.7 **
			50	113.5 ± 10.4 ***
	Desipramine	i.p.	10	93.9 ± 9.7 ***
Rat	Dipraglurant	p.o.	0	214.5 ± 6.1
			3	177.5 ± 5.3 **
			10	184.7 ± 9.2 *
			30	167.5 ± 11.8 **
	Imipramine	i.p.	64	101.8 ± 7.8 ***

**Table 3 cells-12-01004-t003:** Summary of experiments investigating effects of dipraglurant in the rotarod test in mice (administered at 10, 30, 50 mg/kg) and in rats (administered at 3, 10, 30 mg/kg), in comparison with corresponding vehicles, chlordiazepoxide (in mice) or diazepam (in rats). *** *p* < 0.001.

Species	Compound	Administration Route	Dose (mg/kg)	Time on the Rotarod (s) Mean ± S.E.M.
Mouse	Dipraglurant	p.o.	0	144.1 ± 8.6
			10	142.9 ± 7.7
			30	158.3 ± 8.4
			50	137.0 ± 10.6
	Chlordiazepoxide	i.p.	30	8.3 ± 2.5 ***
Rat	Dipraglurant	p.o.	0	103.2 ± 17.5
			3	137.1 ± 21.9
			10	124.2 ± 17.7
			30	102.1 ± 19.6
	Diazepam	i.p.	8	18.7 ± 7.0 ***

## Data Availability

The data are unavailable due to privacy.

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
