# Peer review of "Effect of the Metabotropic Glutamate Receptor Type 5 Negative Allosteric Modulator Dipraglurant on Motor and Non-Motor Symptoms of Parkinson’s Disease"

_cells, 2023, doi:10.3390/cells12071004_

Round 1

Reviewer 1 Report

This study by Epping-Jordan and coworkers investigated at first the pharmacokinetic profile of dipraglurant, a selective mGlu5 negative allosteric modulator (NAM) in rodents. Subsequently, dipraglurant was tested in the haloperidol-induced catalepsy test in rats as well as in several models of anxiety, stress and obsessive-compulsive behavior in rats and/or mice. The authors could show that dipraglurant is rapidly absorbed after oral administration, readily crosses the blood-brain barrier, and displays a positive correlation between plasma concentration and efficacy in behavioral models. Moreover, they showed that dipraglurant reversed haloperidol-induced catalepsy, and exerted anxiolytic-like, antidepressant-like and anticompulsive-like effects, though the latter at higher doses.

This is a straightforward and well conducted study. Methods are sound and the experimental groups for the behavioral tests are large enough to provide adequate statistical power to substantiate the conclusions. The paper is well written and is a very good fit for this Special Issue!

I only have one remark concerning the discussion: the sentences in lines 411-413 are a duplication of those in lines 404-407 and must, therefore, be deleted.

Author Response

Dear Reviewer,

Thank you for your kind comments. The duplicated sentence has been removed.

Best regards,

Robert

Reviewer 2 Report

This is an interesting and relevant work that investigates the role of the mGluR5 receptor in hyperglutamatergism underlying motor and non-motor disorders in Parkinson's disease. My concern is based on why the authors performed the experiments on healthy rodents and not in a model of PD, for example from 6-OHDA. It would have been interesting to evaluate the effect of dipraglurant in Parkinson's-induced anxiety and depression.

Regarding the route of administration, it differs between the dipraglurant (p.o.) and the positive controls (i.p.).

The reason why mice and two strains of rats were used should be given.

Minors

Pain could also be cited among the non-motor symptoms of Parkinson's disease and the action of group II and III mGluRs on it.

The different potency of dipraglurant on different non-motor and species-specific symptoms could also be discussed

Author Response

I would like to thank the reviewer for the insightful comments.

I fully agree with the reviewer that testing anxiety- and depression-like reactivity in a rodent model of PD rather than intact animals would have been very interesting.  However, the classic tests for anxiety- and depression-like reactivity, used in the current study, require intact, unaltered motor activity, which would have been impossible in a PD model, such as 6-OHDA lesion model.  Therefore, we chose intact animals for evaluation of anxiety- and depression-like reactivity. 

Each positive control used in our experiments was chosen from a different class of compounds with the goal to validate the experiment. Therefore, there was no need to use the same route as with dipraglurant. The i.p. route of administration was chosen for positive controls in alignment with their PK profile.  

We used C57Bl/6J mice together with Sprague-Dawley and Wistar rats based on a large body of internal data in corresponding models where the activity of positive and negative controls are robust and well-reproducible.  (added to the text; line 82-84) 

We added pain to the list of non-motor symptoms exhibited by Parkinsonian patients.  Also, we added that mGlu5 NAMs reduce pain in a range of rodent models (Pereira and Goudet 2019); lines 68-69.  

The ED50 values of dipraglurant obtained in the current series of experiments differed across models and species.  For examples, the ED50 of dipraglurant in the rat model of catalepsy (2.83 mg/kg) was markedly lower than the value obtained in the mouse marble burying test (43 mg/kg).  Most likely these are linked to the differences in mGlu5 receptor distribution in neural circuits mediating a particular behavioral response as well as species-specific differences in mGlu5 receptors (Added to the Discussion; lines 485-490)

Reviewer 3 Report

Manuscript entitled “Effect of the metabotropic glutamate receptor type 5 negative 2 allosteric modulator dipraglurant on motor and non-motor 3 symptoms of Parkinson’s disease” is an interesting investigation shedding light on dipraglurant’s protective effect on the non-motor symptoms of PD.

However, I still have some questions for the authors:

Why blood samples are withdrawn after every experiment? Why ED 50 is different in each experiment? And also it is not explained in the discussion section.

Did the authors also see any difference in time spent in center in the OFT test? 

Author Response

The blood samples were withdrawn after ever experiment in order to calculate the concentration of dipraglurant in the plasma and link it with its activity.  

The ED50 values of dipraglurant obtained in the current series of experiments differed across models and species.  For examples, the ED50 of dipraglurant in the rat model of catalepsy (2.83 mg/kg) was markedly lower than the value obtained in the mouse marble burying test (43 mg/kg).  Most likely this is linked to the differences in mGlu5 receptor distribution in neural circuits mediating a particular behavioral response as well as species-specific differences in mGlu5 receptors.

The goal of the locomotor activity experiment with dipraglurant was to confirm lack of sedative-like effects (via measuring the total distance travelled) rather than monitoring anxiety-like reactivity in an Open Field Test.  The equipment used in the experiment, because of its size and lack of brighter illuminated center, was not suitable for the Open Field Test.  However, we fully agree with the reviewer that the Open Field Test would have been another test to show anxiolytic-like profile of dipraglurant.